# Comparison of Genetic Susceptibility to Coronary Heart Disease in the Hungarian Populations: Risk Prediction Models for Coronary Heart Disease

**DOI:** 10.3390/genes14051033

**Published:** 2023-04-30

**Authors:** Nayla Nasr, Beáta Soltész, János Sándor, Róza Ádány, Szilvia Fiatal

**Affiliations:** 1Department of Public Health and Epidemiology, Faculty of Medicine, University of Debrecen, 4032 Debrecen, Hungary; 2Doctoral School of Health Sciences, University of Debrecen, 4032 Debrecen, Hungary; 3Faculty of Public Health, University of Debrecen, 4032 Debrecen, Hungary; 4ELKH-DE Public Health Research Group, Department of Public Health and Epidemiology, Faculty of Medicine, University of Debrecen, 4032 Debrecen, Hungary

**Keywords:** coronary heart disease, developmental models, genetic risk factors, conventional risk factors

## Abstract

**Background and Aim**: It was evaluated whether the integration of genetic risk scores (GRS-unweighted, wGRS-weighted) into conventional risk factor (CRF) models for coronary heart disease or acute myocardial infarction (CHD/AMI) could improve the predictive ability of the models. **Methods**: Subjects and data collected in a previous survey were used to perform regression and ROC curve analyses as well as to examine the role of genetic components. Thirty SNPs were selected, and genotype and phenotype data were available for 558 participants (general: N = 279 and Roma: N = 279). **Results**: The mean GRS (27.27 ± 3.43 vs. 26.68 ± 3.51, *p* = 0.046) and wGRS (3.52 ± 0.68 vs. 3.33 ± 0.62, *p* = 0.001) were significantly higher in the general population. The addition of the wGRS to the CRF model yielded the strongest improvement in discrimination among Roma (from 0.8616 to 0.8674), while the addition of GRS to the CRF model yielded the strongest improvement in discrimination in the general population (from 0.8149 to 0.8160). In addition to that, the Roma individuals were likely to develop CHD/AMI at a younger age than subjects in the general population. **Conclusions**: The combination of the CRFs and genetic components improved the model’s performance and predicted AMI/CHD better than CRFs alone.

## 1. Introduction

Coronary heart disease (CHD) is one of the leading causes of death and morbidity as well as one of the leading causes of premature disability among adult men and women globally [1,2,3]. In 2019, 182 million DALYs were attributed to CHD, where 9.14 million related deaths occurred (nearly 16% of global deaths), and 197 million cases were reported. These numbers increased steadily by 2021 and reached almost 185 million DALYs and 9.44 million deaths [4,5]. CHD also remains the leading cause of death in Hungary; the mortality rate due to CHD was higher than 350 per 100,000 population in 2016, and there were 32,102 deaths (nearly 24.5%) related to CHD in 2020 [6,7,8,9,10]. CHD is a condition of narrowed or blocked arteries that supply the myocardium with oxygen and supplements. The decrease in blood flow and the accumulation of lipids along with the proliferation of smooth muscle cells impair endothelial function [11,12,13]. The development and progression of CHD are a result of the complex interaction of increasing age, male sex, genetic variations (40–60%), and behavioural risk factors such as a sedentary lifestyle, obesity, smoking, harmful alcohol consumption, hyperlipidaemia, hypertension, and diabetes mellitus [14,15,16,17].

Over the past few decades, considerable advances in the clinical diagnosis and curative procedure of CHD have been achieved; however, no significant reduction in the morbidity and mortality from CHD has occurred [18]. The disease burden is increasing steadily, and the DALYs and YLL are also growing substantially. The morbidity and mortality of CHD vary among countries, populations, and specific ethnic minorities, and these variations might be due to large inequalities in socioeconomic status and education, and possibly due to differences in genetic susceptibility [18,19,20,21]. Preventive interventions for CHD are available (medication and lifestyle modification) for high-risk individuals; however, a comprehensive and precise assessment of CHD risk is needed to identify these high-risk subgroups in the early stage of the disease, thus resulting in a more cost-effective treatment strategy [22]. Prediction models play a pivotal role in assessing the risk associated with CHD among those groups, including the Framingham, SCORE, QRISK, and ASSIGN models [18].

The Roma population is a vulnerable and disadvantaged ethnic group in Hungary; most of them suffer from serious deprivation, live in inadequate housing facilities, and are below the poverty line [23,24,25]. Socioeconomic level and health status reflect their socioeconomic status and environmental factors. The Roma population has relatively limited access to healthcare units as a result most Roma CHD patients have a worse cardiometabolic profile at the entry of care, which is characterized by a high risk of premature death [26]. The Roma population is exposed to risk factors for CHD, including heavy smoking, obesity, metabolic syndrome, diabetes mellitus, high triglyceride levels, and low HDL-C concentrations [27,28]. Consequently, an accurate estimation for CHD risk among the Roma population is needed, as no genetic risk prediction study focuses on investigating CHD events in this vulnerable group.

This study aims to predict the risk of CHD among this vulnerable ethnicity group, by estimating and comparing the allele frequencies of 30 selected SNPs associated with this disease, computing and comparing the genetic risk score (weighted and unweighted), and by assessing the role of nongenetic (environmental and behavioral) and conventional risk factors of CHD suggested by the Framingham Score in Hungarian populations (general and Roma). These results might contribute to developing an appropriate intervention model, which might provide a sound opportunity for personalizing prevention strategies to reduce the risk of CHD.

## 2. Materials and Methods

### 2.1. Study Populations

Overall, 558 study subjects were enrolled from two Hungarian populations: the general population (n = 279) and the Roma population (n = 279). They were selected randomly in the framework of a complex comparative health survey. Notably, northeast Hungary is the geographical region where most of the Roma Hungarians live in segregated colonies (Hajdú-Bihar and Szabolcs-Szatmár-Bereg counties). General subjects were chosen from the same source population. Data collection was performed through questionnaires, physical examinations, and laboratory measurements. Blood samples for DNA extraction were also drawn from all the subjects. The details of the sample and data collection have been previously described [29].

### 2.2. Variables Formulated from the Questionnaire Data

Demographic, physical, and laboratory data (i.e., age, sex, BMI, SBP, DBP, TC, HDL-C, LDL-C, TG) were collected via a questionnaire. *CHD and AMI* were confirmed when participants answered “yes” to one of the following questions: “Did you have CHD or AMI during the last 12 months?”, “Have you been diagnosed with CHD or AMI by a medical doctor?”, or “Have you received hospital treatment for CHD or AMI?”. *Hypertension* was confirmed when participants answered “yes” to one of the following questions: “Did you have HTN during the last 12 months?”, “Have you been diagnosed with HTN by a medical doctor?”, or “Have you received a hospital treatment for HTN?” *Elevated hypertension* was defined as an average SBP ≥ 140 mmHg, or a DBP ≥ 90 mmHg based on the Fifth Joint National Committee Guideline (JNC-V) [30]. In addition, we defined hypertension based on the current use of hypertensive medication and an answer of “yes” to the following question: “Have you received a hospital treatment for hypertension during the last 12 months?”. The *lipid profiles* (HDL-C, LDL-C, and TG) were determined in the laboratory (mmol/L), while *elevated TC* was confirmed when participants answered “yes” to one of the following questions: “Did you have a high cholesterol level during the last 12 months?”, “Have you been diagnosed with a high cholesterol level by a doctor?”, or “Have you received a hospital treatment for a high cholesterol level?” *Diabetes mellitus* (DM) was confirmed when participants answered “yes” to one of the following questions: “Did you have a DM during the last 12 months?”, “Have you been diagnosed with DM by the medical doctor?”, or “Have you received a hospital treatment for DM?”. *Stroke* was confirmed when participants answered “yes” to one of the following questions: “Did you have a stroke during the last 10 months?”, “Have you been diagnosed with a stroke by a medical doctor?”, or “Have you received a hospital treatment for stroke?”. *Chronic kidney disease* (CKD) was confirmed when participants answered “yes” to one of the following questions: “Did you have CKD during the last 12 months?”, “Have you been diagnosed with CKD by a medical doctor?”, or “Have you received hospital treatment for CKD?” *Smoking* status was confirmed when participants answered “yes” to the following question: “Are you a current smoker?”

### 2.3. SNPs Selection Procedure and Genotyping

Genetic variants (30 SNPs) associated with CHD were selected from previous studies (see Appendix A). In general, twenty-five SNPs that were significantly associated with CHD at a genome-wide level in prior analyses (see the details in Mega et al.) were selected [31]. Furthermore, an additional three variants were added based on a paper by Tikkanen et al. [32], and two SNPs were selected from Schenkert et al. [33] and Teslovich et al. [34]. These publications that had several overlapping SNPs were the most cited GWASs on CHD. Genotyping by the Mutation Analysis Facility, Clinical Research Centre, Karolinska University Hospital (Sweden) was successfully performed for all SNPs.

### 2.4. Weighted and Unweighted Genetic Risk Score (GRS, wGRS) Constructions

SNPs were coded based on the number of risk alleles as follows: 0 was assigned in the absence of the risk allele, subjects homozygous for the risk allele were coded as 2, and the single risk allele was coded as 1. The unweighted GRS was calculated by simply counting the number of risk alleles present for each SNP for every study subject, while the wGRS was computed by multiplying the risk allele score (0, 1, 2) carried for each SNP by the published effect size estimate (ln of odds ratio) [35,36].

### 2.5. Statistical Analysis and Software Used

First, DNA samples and the questionnaire data were matched; individuals with any missing genotype and phenotype values or individuals who did not specify their gender were excluded before the statistical analysis to minimize the possibility of systematic errors. Data quality control guidelines were applied based on a previous publication [37,38].

The allele frequencies of 30 SNPs associated with CHD in the Hungarian general and Hungarian Roma populations were analysed using Plink 1.07 software. The force-specific allele technique was used to ensure that the affected risk alleles were assigned to be first before running the commands in the allele frequency comparison analyses [39]. The HWE test in Plink 1.07 was used to exclude any SNPs that failed to be in HWE [40]. The LD calculation was performed using Haploview 4.2 software to examine whether there was any correlation between these SNPs [41] (see Appendix A). Bonferroni correction was performed (α new = (α old/n) (*p*-value 0.05/29 = 0.002) [42]. Two-sided t-tests and Pearson’s chi-square tests were used to test the differences in the means and the proportions of the predictors between the study groups. Binary logistic regression analysis was conducted by using Stata 13 software to assess the relationship between the weighted and unweighted GRSs as independent predictors for CHD risk when integrated with the conventional risk factors.

We created different models by using variables (age, sex, total cholesterol level, SBP, and smoking as conventional risk factors) suggested by the SCORE model [43,44]. In addition, we updated the model by adding some potential explanatory predictors, such as genetic risk scores (wGRS and GRS divided into tertiles for low-, intermediate-, and high-risk subgroups) and diabetes mellitus (DM). Hypertension (elevated blood pressure and/or taking blood pressure-lowering therapy), high cholesterol levels (elevated TC and/or taking cholesterol-lowering therapy), and ethnicity were also included in the statistical models, as shown in Table 1.

To evaluate the potential value of the integration of CRFs and genetic risk score (GRS and wGRS) in risk prediction, the model’s performance (discrimination, calibration) was assessed. First, the area under the receiver operating characteristic curve of models with and without the GRS (GRS and wGRS) was computed. Second, the calibration by the Hosmer–Lemeshow goodness-of-fit test was measured. Marginal plot analysis was also used to predict the interaction between the genetic risk score (GRS, wGRS), age, and sex for CHD/AMI risk prediction.

## 3. Results

### 3.1. Demographic Characteristics of the Study Population

In general, the complete genotype and phenotype information was available for 558 participants: 279 from the Hungarian Roma population and 279 from the Hungarian general population. The mean (SD) age was 42.73 ± 12.99 years for the Hungarian Roma population and 44.14 ± 12.12 years for the Hungarian general population. The results clarified a significant difference between sexes, TC-Med, smoking status, and prevalence cases of DM, CKD, and HTN-Med. The prevalence rates of CHD and AMI were low among both groups (Roma and general), and there was no significant difference between groups. The results also indicated that the Hungarian Roma population was younger, had a lower proportion of male subjects, lower HDL-C levels, and lower mean SBP (123.57 ± 17.73). Furthermore, they had a greater tendency to be a current smoker and were likely to be exposed to chronic diseases, including DM, HTN, and CKD. Moreover, the prevalence of CHD, AMI, and stroke was higher among the Roma group (see Table 2).

### 3.2. Frequencies and Associations of the Individual Genetic Variants Related to CHD Risk

After the HWE test, one SNP (rs12413409) showed deviation in the Hungarian Roma population and thus was excluded from further GRS computation (see Appendix A). The allele frequencies of 30 SNPs associated with CHD risk in the Hungarian (general and Roma) population are shown in Table 3. These results were consistent with the results from previously published evidence (see Appendix A).

It was revealed that nine SNPs had significantly different prevalence in the two study populations even after the Bonferroni correction: six SNPs, rs2306374 (gene *MRAS*), rs9818870 (gene *MRAS*), rs12190287 (gene *TCF21*), rs10455872 (gene *LPA*), rs3184504 (gene *SH2B3*), and rs9982601 (gene KCNE2), were more prevalent in the Hungarian general population, while three SNPs, rs17609940 (gene *ANKS1A*), rs2259816 (gene *HNF1A*), and rs12936587 (gene *RASD1*), occurred more frequently among Roma individuals.

It was revealed (Figure 1 and Figure 2) that the mean GRS and wGRS were higher in the general population than in the Roma population (27.27 ± 3.43 vs. 26.68 ± 3.51, *p*-value = 0.046 and 3.52 ± 0.68 vs. 3.33 ± 0.62, *p*-value = 0.001, respectively).

### 3.3. Multivariable Regression Analyses for CHD/AMI

The odds ratio of CRFs according to SCORE-based models for CHD/AMI risk prediction revealed that age and elevated cholesterol or therapy for high cholesterol level were associated significantly with CHD/AMI risk in the Hungarian general population ((OR = 1.183, *p*-value = 0.046, 95% CI 1.001–1.172) and (OR = 4.899, *p*-value = 0.032, 95% CI 0.236–0.219)), respectively, while HTN-Med showed a significant association with CHD/AMI risk only among the Roma (OR = 7.849, *p*-value = 0.001, 95% CI 2.412–25.547) population (Table 4).

When we combined the study population and used ethnicity as a possible predictor: age (OR = 1.050, *p*-value = 0.030, 95% CI 1.005–1.096), HTC (OR = 3.553, *p*-value = 0.007, 95% CI 1.418–8.901), and HTN-Med (OR = 4.790, *p*-value = 0.001, 95% CI, 1.902–12.066) were associated significantly with CHD/AMI in the Hungarian populations (Table 5). Although ethnicity did not prove to be a significant predictor, Roma seems to have a higher risk of developing CHD/AMI independently from conventional risk factors.

Although no significant association was observed between CHD/AMI and GRS/wGRS in these models, the ORs revealed a proportionally increased risk in the second and third tertiles in the general population (Table 6 and Table 7).

The addition of GRS and wGRS tertiles to CRFs revealed that age, HTC-Med, and HTN-Med were significantly associated with CHD/AMI in the combined population independent of the effect of ethnicity (Table 8 and Table 9). The results of the analyses for each group are shown in the Appendix A (Appendix A). Although ethnicity was not shown to be a significant and independent risk factor by itself, the results suggest that the Roma population has a higher risk of developing AMI/CHD.

By including DM as a reasonable new predictor to the updated SCORE-based genetic models (GRS and wGRS included), age, HTC-Med, and HTN-Med remained significant risk factors for developing CHD/AMI (for more details, see Table 10, Table 11 and Table 12, Appendix A). However, the GRSs still did not show a significant association with CHD/AMI.

### 3.4. ROC Curve Analyses

The models’ performances (discrimination, calibration, and risk classification) are described in Table 13 and Table 14. The AUC curve estimates LROC for CHD/AMI risk prediction when integrating the CRF basic, CRF with GRS, and CRF with wGRS models, which were 0.8149, 0.8346, and 0.8160 in the Hungarian general population and 0.8616, 0.8549, and 0.8674 in the Roma population, respectively. When DM was added to the models, minimal improvement occurred in the AUC values in the general population, and no significant improvement was observed among Roma individuals (Table 13). Considerable improvements in the AUC value were observed when the CRF basic model with DM was combined with GRS in the Hungarian general population (from 0.8299 to 0.8400). For the Roma population, the CRF basic model (without DM) showed the greatest improvements in the AUC value (0.8616 to 0.8674) compared to the CRFS+wGRS model. In the combined populations (Table 14), the CRF basic model with DM and ethnicity showed the greatest AUC value (LROC = 0.8525) compared to the basic model (without DM). The results of the Hosmer–Lemeshow tests indicated that all the models had a good fit (*p*-value ≥ 0.11).

The greatest improvement in the AUC values among the models occurred when we added the unweighted GRS to the conventional risk factors (See Appendix A).

### 3.5. Marginal Plot Analyses

In the Hungarian general population, CHD/AMI risk was low among males (margin = 0.001, *p* = 0.707; 95% CI: −0,001–0.001) and females (margin = 0.006, *p* = 0.516; 95% CI: −0.013–0.026) subjects between 18 and 45 years of age. The risk of the trait increased after the age of 46 years among females (margin = 0.028, *p*-value = 0.048, 95% CI 0.000–0.056) and after 54 years among male subjects (margin = 0.065, *p*-value = 0.039, 95% CI 0.003–0.126); thus, female subjects were predicted to develop CHD/AMI earlier than male subjects. For the Roma population, the marginal plot shows that CHD/AMI risk is low for younger subjects (males and females between 18 and 40 years of age; margin = 0.03, *p* = 0.397; 95% CI: −0.043–0.109 and margin = 0.003, *p* = 0.387; 95% CI: −0.004–0.011, respectively). The risk starts to significantly increase at the age of 41 years for both males and females (male subjects: margin = 0.080, *p*-value = 0.047, 95% CI: 0.001–0.159; female subjects: margin = 0.024, *p*-value = 0.045, 95% CI 0.000–0.048 for age 41 years) (see Appendix A).

The prediction of CHD/AMI risk in the combined population revealed that the risk is higher for the Roma population. The risk becomes significant at 34 years of age in the Roma group (margin = 0.022, *p*-value = 0.046, 95% CI: 0.000–0.043), while for the general population, the risk becomes significant at the age of 44 (margin = 0.020, *p*-value = 0.046, 95% CI: 0.000–0.041) (see Appendix A).

## 4. Discussion

The Hungarian population, especially the Roma living in this country, has a high risk for cardiometabolic syndrome [21,26,45,46,47,48,49]; thus, it was hypothesized that this ethnicity is more susceptible to developing CHD than the general population. A cross-sectional study-based population from Hungary was used to examine whether the integration of unweighted and weighted genetic risk scores computed from 29 SNPs into a SCORE-based prediction model improves the risk prediction of CHD/AMI. Our study revealed that the mean weighted GRS and GRS were higher for the general population. Although the vast majority of the selected SNPs were not independently associated with CHD/AMI, it seems that the general population is more likely to develop CHD/AMI.

Age is an essential non-modifiable risk factor for CHD/AMI [1]. Age was associated with CHD/AMI development in the multivariable regression model, especially in the general population. The marginal plot analyses revealed that CHD/AMI risk is obviously low among the younger populations generally; however, both populations (general and Roma) were susceptible to premature onset of CHD/AMI (before age 55 years), and the risk was expected to begin earlier among Roma individuals.

Male sex is an independent predictor for CHD (especially for AMI mortality) and plays a role in advancing CHD risk, although the mortality rate of CHD is still higher among women than men [4,10,21,50,51,52,53,54,55]. Marginal plot predictions revealed that the risk of CHD/AMI increased among female subjects of the Hungarian (general and Roma) population compared to male subjects. However, Roma (male and female) individuals tend to develop CHD/AMI before the Hungarian general population.

Hyperlipidaemia (elevated total cholesterol or lipid-lowering medication) was a major risk factor for CHD/AMI in the general population, and hypertension (elevated blood pressure or antihypertensive medication) showed a significant association with the trait only in the Roma population. DM showed no association with CHD risk in all models, possibly due to the low prevalence.

Smoking is known to be the strongest contributor to CHD/AMI in all available models, such as the Framingham, SCORE, QRISK1, QRISK2, and PROCAM models [11,12,18,43,55,56,57,58], all of which have been developed and validated to predict CHD risk in general populations. Herein, smoking was a nonsignificant predictor of an increased risk of CHD in the Hungarian populations, which might be due to selection bias or interaction as a confounding factor with some other predictors in the Hungarian populations. One reason for this could be the presence of other highly correlated predictor variables in the model that have a stronger association with cardiovascular diseases. For example, high blood pressure, high cholesterol levels, and obesity are also well-established risk factors for cardiovascular diseases and may have a stronger association with the outcome variable in the multiple regression model. In this case, the effect of smoking may be overshadowed by the effects of these other variables, thus making smoking a nonsignificant factor.

Based on previous studies [2,59,60,61,62,63,64], the combination of the CRFs (based on SCORE variables) and genetic components (GRS or wGRS) might improve the model performance and predict AMI/CHD better than CRFs alone. Although GRS/wGRS were not significantly associated with CHD/AMI risk in study populations, the utility of genetic factors is unquestionable in risk prediction. We examined the predictive ability of the combined models where CRFs (basic model) and genetic risk scores were integrated. A basic model that included DM showed good discrimination improvement compared to the model without DM. In addition, the models with GRS integration showed the greatest discrimination improvement in general.

With a careful selection of CHD/AMI-associated SNPs, we tried to create a genetic scoring model that was able to provide a somewhat valid estimate (made possible by the chosen method) of the genetic load in our study populations; however, due to the small number of CHD/AMI patients in the study groups, the GRSs were not significantly associated with the trait in the regression models, but the predictive accuracy of the models having a genetic component showed a remarkable AUC improvement. In the future, validating the genetic risk prediction model using independent datasets or cohorts to assess its accuracy, reliability, and generalizability would help to determine the robustness and external validity of our model. An evaluation of the clinical utility and potential impact of the genetic risk prediction model in clinical settings would also be important. This may involve conducting prospective studies or implementation trials to assess the effectiveness of our model in clinical practice, and its potential for guiding personalized risk assessment, prevention, and treatment strategies.

Nevertheless, the utilization of genetic risk prediction models has the potential to improve health outcomes by providing personalized risk estimates and aiding in early detection and intervention. However, careful consideration and communication of the limitations and challenges of these models are necessary for their effective implementation in clinical practice.

## Figures and Tables

**Figure 1 genes-14-01033-f001:**
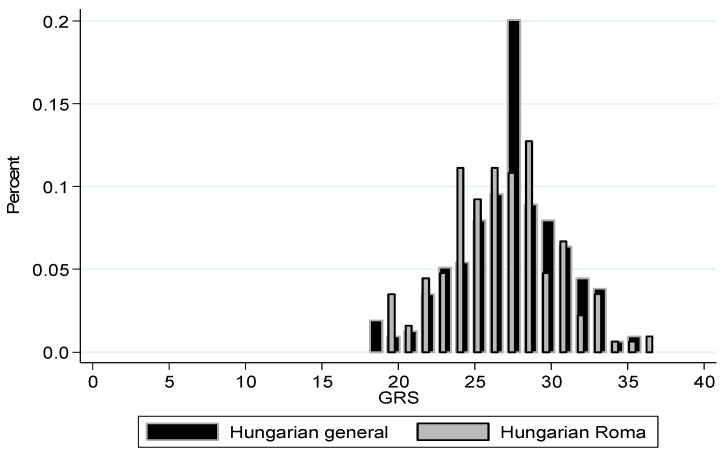
The unweighted genetic risk score distribution among the Hungarian general and Hungarian Roma populations (*p*-value = 0.046).

**Figure 2 genes-14-01033-f002:**
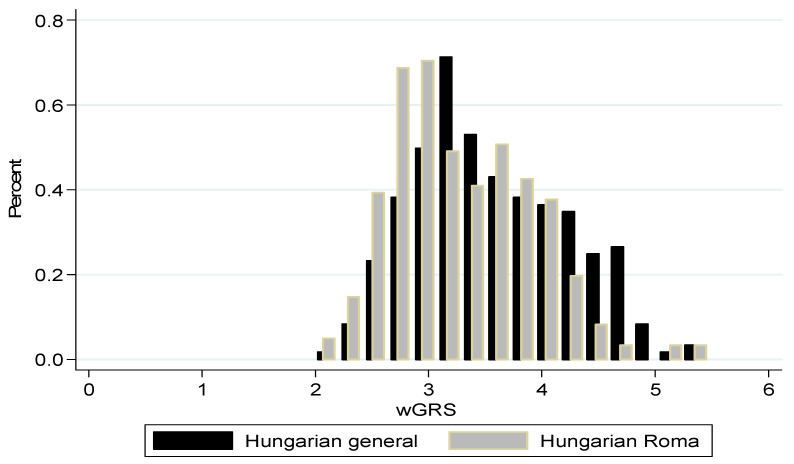
Distribution of the weighted genetic risk score in study populations (*p*-value = 0.001).

**Table 1 genes-14-01033-t001:** Multivariable logistic models developed for CHD and AMI risk prediction.

Models	Explanatory Variables
SCORE based models + ethnicity	age, sex, HTC-Med, HTN-Med, smoking (CRFs)
	CRFs + ethnicity *
Genetic based models only	GRS per tertiles
	wGRS per tertiles
SCORE based models + genetic models + ethnicity	CRFs + GRS + ethnicity
	CRFs + wGRS + ethnicity
SCORE based models upgraded *	CRFs + DM
	CRFs + DM + ethnicity
SCORE based models + genetic models + ethnicity upgraded *	CRFs + DM + GRS + ethnicity
	CRFs + DM + wGRS + ethnicity

* The Hungarian population was set as the reference. Outcome variable: CHD/AMI. TC-Med: high total cholesterol level/or taking cholesterol-lowering therapy; HTN-Med: elevated blood pressure and/or taking blood pressure lowering therapy; CRFs: conventional risk factors; GRS: unweighted GRS; wGRS: weighted GRS. * Upgraded by adding DM as an explanatory variable.

**Table 2 genes-14-01033-t002:** Demographic characteristics of the Hungarian general and Roma populations.

Characteristics	Hungarian General (n = 279)	Hungarian Roma (n = 279)	*p*-Value
Age (years, mean ± SD)	44.14 ± 12.12	42.73 ± 12.99	0.177
18–36 age-group (%)	27.60	34.77	
37–49 age-group (%)	35.48	30.82	
50–77 age-group (%)	36.92	34.41	
Sex male/female (%)	**43.73/56.27**	**23.3/76.7**	**<0.001**
BMI kg/m^2^ (mean ± SD)	27.11 ± 5.49	27.29 ± 6.72	0.725
SBP mmHg (mean ± SD)	126.94 ± 15.23	123.57 ± 17.73	0.017
DBP mmHg (mean ± SD)	78.91 ± 8.71	79.97 ± 10.26	0.189
TC mmol/l (mean ± SD)	4.95 ±1.14	4. 90 ± 1.07	0.563
HDL-C mmol/l (mean ± SD)	**70.11 ± 33.15**	**61.00 ± 30.98**	**0.001**
LDL-C mmol/l (mean ± SD)	3.07 ± 1.00	3.15 ± 0.94	0.328
TG mmol/l (mean ± SD)	1.58 ± 1.09	1.57 ± 0.95	0.884
CHD yes/no (%)	2.87/97.13	3.94/9.06	0.484
AMI yes/no (%)	1.08/98.92	3.23/96.77	0.080
Stroke yes/no (%)	1.43/98.57	3.23/96.77	0.161
DM yes/no (%)	**6.09/93.91**	**12.54/87.46**	**0.009**
HTN yes/no (%)	28.32/71.68	34.05/65.95	0.144
CKD yes/no (%)	0.72/99.28	4.3 /95.7	0.007
HTN-Med yes/no (%)	**4.3/95.7**	**15.41/84.59**	**<0.001**
HTC-Med yes/no (%)	7.89/ 92.11	14.34/ 85.66	0.015
Smoking currently yes/no (%)	**32.97/67.03**	**69.95/34.05**	**<0.001**

BMI: body mass index, SBP: systolic blood pressure, DBP: diastolic blood pressure, TC: total cholesterol, HDL-C: high-density lipoprotein cholesterol, LDL-C: low-density lipoprotein cholesterol, TG: triglyceride, CHD: coronary heart disease, AMI: acute myocardial infarction, DM: diabetes mellitus, HTN: hypertension, CKD: chronic kidney disease, HTN-Med: anti-hypertensive medication, HTC-Med: medication against high total cholesterol. (Bold values indicate significant difference).

**Table 3 genes-14-01033-t003:** CHD risk allele frequencies in Hungarian General and Roma Hungarian populations.

rs Number	CHR	RA	Roma	Hungarian	OA	*p*-Value
rs11206510	1	T	0.875	0.814	C	0.005
rs17114036	1	A	0.919	0.918	G	0.913
rs646776	1	T	0.823	0.789	C	0.151
rs17465637	1	C	0.654	0.703	A	0.084
rs6725887	2	C	0.084	0.100	T	0.352
rs2306374	3	C	0.066	**0.145**	T	**<0.001**
rs9818870	3	T	0.066	**0.142**	C	**<0.001**
rs9349379	6	G	0.475	0.430	A	0.133
rs12526453	6	G	0.339	0.344	C	0.850
rs17609940	6	G	**0.952**	0.884	C	**<0.001**
rs12190287	6	C	0.552	**0.647**	G	**0.001**
rs3798220	6	C	0.002	0.013	T	0.033
rs10455872	6	G	0.020	**0.057**	A	**0.001**
rs11556924	7	C	0.706	0.669	T	0.175
rs4977574	9	G	0.570	0.488	A	0.006
rs579459	9	C	0.238	0.285	T	0.077
rs635634	9	T	0.192	0.262	C	0.005
rs1746048	10	C	0.792	0.817	T	0.291
rs12413409	10	G	0.801	0.862	A	0.007
rs964184	11	G	0.194	0.145	C	0.031
rs3184504	12	T	0.366	**0.529**	C	**<0.001**
rs2259816	12	T	**0.480**	0.344	G	**<0.001**
rs4773144	13	G	0.414	0.416	A	0.952
rs2895811	14	C	0.344	0.412	T	0.019
rs3825807	15	A	0.504	0.590	G	0.004
rs216172	17	C	0.285	0.324	G	0.153
rs12936587	17	G	**0.731**	0.597	A	**<0.001**
rs46522	17	T	0.572	0.548	C	0.433
rs1122608	19	G	0.720	0.769	T	0.064
rs9982601	21	T	0.086	**0.147**	C	**0.002**

CHR: chromosome, RA: Risk allele, OA: another allele. Bold values indicate those allele frequencies that remain significant after the Bonferroni correction.

**Table 4 genes-14-01033-t004:** The odds ratio associated with CRFs in the model for predicting CHD/AMI risk in the Hungarian populations.

CHD/AMI	Hungarian General			Hungarian Roma		
	OR	*p*-Value	95% CI	OR	*p*-Value	95% CI
Age	1.183	0.046 *	1.001–1.172	1.029	0.317	0.973–1.087
Sex(Male)	1.623	0.480	0.424–6.155	2.338	0.149	0.738–7.403
HTC-Med	4.899	0.032 *	0.236–0.219	2.999	0.078	0.885–10.157
HTN-Med	1.373	0.781	1.151–20.848	7.849	0.001 *	2.412–25.547
Smoking	0.916	0.905	0.147–12.778	1.264	0.704	0.377–4.232

* Statistically significant (*p*-value < 0.05).

**Table 5 genes-14-01033-t005:** Odds ratio associated with CRFs in model for predicting CHD/AMI risk in the combined populations.

CHDAMI	OR	*p*-Value	95% CI
Age	1.050	0.030 *	1.005–1.096
Sex (Male)	1.916	0.139	0.809–4.537
Ethnicity **	1.468	0.410	0.588–3.665
HTC-Med	3.553	0.007 *	1.418–8.901
HTN-Med	4.790	0.001 *	1.902–12.066
Smoking	1.059	0.901	0.427–2.628

** *Hungarian general population was a reference population.* * *Statistically significant (p-value < 0.05)*.

**Table 6 genes-14-01033-t006:** Odds ratio of the GRS-based model for CHD/AMI in the Hungarian population.

CHD/AMI	Hungarian General			Hungarian Roma		
	OR	*p*-Value	95% CI	OR	*p*-Value	95% CI
GRS-T2	1.965	0.550	0.215–17.993	0.372	0.125	0.105–1.317
GRS-T3	2.828	0.348	0.322–24.818	0.668	0.508	0.203–2.020

GRS tertiles; GRS T1 (18–24) was set as reference, GRS T2 (25–28), GRS T3 (29–37).

**Table 7 genes-14-01033-t007:** Odds ratio of the wGRS based model for CHD/AMI in the Hungarian populations.

CHD/AMI	Hungarian General			Hungarian Roma		
	OR	*p*-Value	95% CI	OR	*p*-Value	95% CI
wGRS-T2	1.113	0.908	0.181–6.837	0.926	0.899	0.283–3.024
wGRS-T3	1.800	0.490	0.340–9.539	0.779	0.699	0.220–2.756

wGRS tertiles; wGRS T1 (1.964–3.047) was set as reference, wGRS T2 (3.049–3.686), wGRS T3 (3.692–5.509).

**Table 8 genes-14-01033-t008:** Odds ratio of CRFs plus ethnicity and GRS for CHD/AMI risk prediction in the study populations.

CHDAMI	OR	*p*-Value	95% CI
Age	1.047	0.040 *	1.002–1.094
Sex(Male)	2.056	0.114	0.841–5.022
Ethnicity **	1.473	0.411	0.586–3.706
HTC-Med	3.862	0.005 *	1.513–9.855
HTN_Med	4.674	0.001 *	1.820–12.007
Smoking	1.026	0.957	0.409–2.573
GRS-T2	0.528	0.261	0.173–1.610
GRS-T3	0.995	0.993	0.338–2.928

** *Hungarian general population was a reference population.* * *Statistically significant (p-value < 0.05)*.

**Table 9 genes-14-01033-t009:** Ethnicity and wGRS for predicting CHDAMI risk in the study populations.

CHDAMI	OR	*p*-Value	95% CI
Age	1.051	0.029 *	1.005–1.098
Sex(Male)	1.879	0.161	0.778–4.539
Ethnicity **	1.465	0.413	0.587–3.660
HTC-Med	3.524	0.007 *	1.406–8.832
HTN_Med	4.907	0.001 *	1.920–12.541
Smoking	1.058	0.903	0.426–2.630
wGRS-T2	1.169	0.779	0.393–3.484
wGRS-T3	1.067	0.904	0.371–3.069

** *Hungarian general population was a reference population.* * *Statistically significant (p-value < 0.05), and wGRS-T1 were set as references.*

**Table 10 genes-14-01033-t010:** Odds ratio of CRFs with DM for CHD/AMI risk prediction among the study populations.

CHDAMI	OR	*p*-Value	95% CI
Age	1.046	0.046 *	1.001–1.094
Sex(Male)	1.942	0.134	0.816–4.621
Ethnicity **	1.559	0.346	0.619–3.928
HTC-Med	3.322	0.011 *	1.315–8.391
DM	1.677	0.323	0.602–4.678
HTN_Med	4.406	0.002 *	1.711–11.342
Smoking	1.053	0.911	0.424–2.620

** *Hungarian general population was a reference population.* * *Statistically significant (p-value < 0.05).*

**Table 11 genes-14-01033-t011:** Odds ratio of CRFs plus ethnicity, DM, and GRS for CHDAMI risk prediction model among the study populations.

CHDAMI	OR	*p*-Value	95% CI
Age	1.044	0.059	0.998–1.091
Sex (Male)	2.061	0.115	0.839–5.063
Ethnicity **	1.545	0.359	0.609–3.916
HTC-Med	3.605	0.008 *	1.399–9.289
DM	1.572	0.395	0554–4.458
HTN-Med	4.395	0.002 *	1.684–11.458
Smoking	1.025	0.958	0.408–2.576
GRS-T2	0.559	0.311	0.181–1.724
GRS-T3	1.030	0.958	0.347–3.056

** *Hungarian general population was a reference population.* * *Statistically significant (p-value < 0.05).*

**Table 12 genes-14-01033-t012:** Odds ratio of CRFs plus ethnicity, DM, and wGRS for CHDAMI risk prediction model among the study populations.

CHDAMI	OR	*p*-Value	95% CI
Age	1.047	0.044 *	1.001–1.095
Sex (Male)	1.888	0.161	0.776–4.591
Ethnicity **	1.558	0.347	0.618–3.926
HTC-Med	3.276	0.012 *	1.296–8.280
DM	1.711	0.306	0.612–4.783
HTN_Med	4.561	0.002*	1.747–11.908
Smoking	1.051	0.914	0.422–2.618
wGRS-T2	1.242	0.701	0.411–3.753
wGRS-T3	1.104	0.857	0.378–3.223

** *Hungarian general population was a reference population.* * *Statistically significant (p-value < 0.05).*

**Table 13 genes-14-01033-t013:** Models’ performances for CHD/AMI risk based on the CRFs and GRS/wGRS in the Hungarian populations.

Models	Hungarian General					Hungarian Roma				
	SENS	SPEC	CLASS	CALIB	LROC	SENS	SPEC	CLASS	CALIB	LROC
1. CHD/AMI models basic SCORE										
CRFs basic	60.00	82.90	82.08	0.9601	0.8149	81.25	76.81	77.06	0.2193	0.8616
**CRFs+GRS**	70.00	85.50	84.95	0.9241	**0.8346**	81.25	77.57	77.78	0.7087	**0.8549**
**CRFs+wGRS**	60.00	84.01	83.15	0.9349	**0.8160**	81.25	77.57	77.78	0.3803	**0.8674**
2. CHD/AMI models based on the updated SCORE (plus DM)										
**CRFs+DM**	60.00	86.99	86.02	0.2818	**0.8299**	81.25	77.19	77.42	0.2298	**0.8611**
**CRFs+DM+GRS**	60.00	86.99	86.02	0.6587	**0.8400**	81.25	77.57	77.78	0.5061	**0.8534**
**CRFs+DM+wGRS**	60.00	86.62	85.66	0.9496	**0.8333**	81.25	77.19	77.42	0.3807	**0.8670**

CFRs: conventional risk factor, GRS: unweighted GRS, wGRS: weighted GRS, SENS: sensitivity, SPEC: specificity, CLASS: Classification, CALIB: calibration, LROC: logit ROC measures. The bold highlight indicated that the LROC is showing improvement after adding GRS, wGRS, and DM to the basic model.

**Table 14 genes-14-01033-t014:** Models’ performances for CHD/AMI risk based on the CRFs and GRS/wGRS in the combined populations using ethnicity predictor.

Models	Hungarian General				
	SENS	SPEC	CLASS	CALIB	LROC
1. CHD/AMI models basic SCORE					
CRFs+E	76.92	81.02	80.82	0.7843	0.8479
CRFs+GRS+E	80.77	81.77	81.18	0.1082	**0.8490**
CRFs+wGRS+E	76.92	81.02	80.82	0.7922	0.8456
2. CHD/AMI models based on the updated SCORE					
CRFs+DM+E	80.77	80.83	80.82	0.5329	**0.8525**
CRFs++DM+GRS+E	80.77	80.83	80.82	0.4737	0.8518
CRFs+DM+wGRS+E	80.77	81.20	81.18	0.7842	0.8497

E: Ethnicity, CFRs: conventional risk factor, GRS: unweighted GRS, wGRS: weighted GRS, SENS: sensitivity, SPEC: specificity, CLASS: Classification, CALIB: calibration, LROC: logit ROC measures. The bold highlight indicated that the LROC is showing improvement after adding Ethnicity (E), GRS, and DM to the basic model.

## Data Availability

All necessary data related to this study will be made available by the authors without undue reservation.

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
