# Peer review of "Comparison of Genetic Susceptibility to Coronary Heart Disease in the Hungarian Populations: Risk Prediction Models for Coronary Heart Disease"

_genes, 2023, doi:10.3390/genes14051033_

Round 1

Reviewer 1 Report

This study evaluated the influence of incorporating the unweighted or weighted genetic risk scores with the traditional risk factor models for investigating the coronary heart disease or acute myocardial infarction. The study is very well performed and results look promising to further expand such studies. 

Please provide additional explanation for the following:

1. How did you select the publications from which you selected 30 SNPs of interest? What were the selection criteria?

2. Line 126 - what was the "published effect size"? For situations with multiple data sources for the same SNP did you took into account the average effect size from all these publications or? 

3. Why have you included GRS and wGRS into multivariate regression models only as tertiles? Have you tried including it into the model as a continuous variable?

Minor comments:

1. Please address all abbreviations when first mentioned in full text (e.g. line 83).

2. Supplementary Table 1 - please provide description of RAF 1/2/3.

3. Sentence in lines 194-195 is repeating (already stated in the paragraph before the Table 3)

4. Table 3 - define that bold p-values represent those that remained significant after Bonferroni correction

5. In tables with results of multivariate regression please define ethnicity (what is referent value)

Reviewer 2 Report

 I think the paper touches a good topic and addresses an interesting question, and I consider that this study is worthy to be published after addressing the following points:

1. Abstract can be reorganized. Background, aim, method, results, conclusion etc.

2. Both motivations and contributions are unclear in Introduction.

3. High-quality figures are suggested.

4. Literature gaps should be given.

5. A comparative study was not included enough. Hence, there appears to be little basis for concluding that the proposed method is more reliable than other methods.

6. More details should be given in Future works.

7.  Some grammatical and punctuation errors were found in the manuscript. Authors are therefore urged to have the entire manuscript reviewed and correct by a native English speaker.
